# Cardiovascular Manifestation of the BNT162b2 mRNA COVID-19 Vaccine in Adolescents

**DOI:** 10.3390/tropicalmed7080196

**Published:** 2022-08-19

**Authors:** Suyanee Mansanguan, Prakaykaew Charunwatthana, Watcharapong Piyaphanee, Wilanee Dechkhajorn, Akkapon Poolcharoen, Chayasin Mansanguan

**Affiliations:** 1Bhumibol Adulyadej Hospital, Bangkok 10220, Thailand; 2Department of Clinical Tropical Medicine, Faculty of Tropical Medicine, Mahidol University, Bangkok 10400, Thailand; 3Department of Tropical Pathology, Faculty of Tropical Medicine, Mahidol University, Bangkok 10400, Thailand; 4Samitivej Srinakarin Hospital, Bangkok 10250, Thailand

**Keywords:** BNT162b2 mRNA COVID-19 vaccine, COVID-19 vaccine, cardiovascular manifestation, myocarditis, adolescents, Thailand

## Abstract

This study focuses on cardiovascular manifestation, particularly myocarditis and pericarditis events, after BNT162b2 mRNA COVID-19 vaccine injection in Thai adolescents. This prospective cohort study enrolled students aged 13–18 years from two schools, who received the second dose of the BNT162b2 mRNA COVID-19 vaccine. Data including demographics, symptoms, vital signs, ECG, echocardiography, and cardiac enzymes were collected at baseline, Day 3, Day 7, and Day 14 (optional) using case record forms. We enrolled 314 participants; of these, 13 participants were lost to follow-up, leaving 301 participants for analysis. The most common cardiovascular signs and symptoms were tachycardia (7.64%), shortness of breath (6.64%), palpitation (4.32%), chest pain (4.32%), and hypertension (3.99%). One participant could have more than one sign and/or symptom. Seven participants (2.33%) exhibited at least one elevated cardiac biomarker or positive lab assessments. Cardiovascular manifestations were found in 29.24% of patients, ranging from tachycardia or palpitation to myopericarditis. Myopericarditis was confirmed in one patient after vaccination. Two patients had suspected pericarditis and four patients had suspected subclinical myocarditis. In conclusion, Cardiovascular manifestation in adolescents after BNT162b2 mRNA COVID-19 vaccination included tachycardia, palpitation, and myopericarditis. The clinical presentation of myopericarditis after vaccination was usually mild and temporary, with all cases fully recovering within 14 days. Hence, adolescents receiving mRNA vaccines should be monitored for cardiovascular side effects. Clinical Trial Registration: NCT05288231.

## 1. Introduction

In December 2020, the US Food and Drug Administration (FDA) issued an Emergency Use Authorization (EUA) for the Pfizer–BioNTech mRNA vaccine (BNT162b2) for the prevention of COVID-19 disease. Clinical trials have revealed that the vaccine’s efficacy is 95% and its safety profile is good, similar to that of other vaccines [1,2,3,4]. Systemic reactions to the vaccine, which were usually mild and transient, have been reported more commonly among the younger population and more often after the second dose [1,2,5].

Historically, post-vaccination myocarditis has been reported as a rare adverse event after vaccinations, especially smallpox [4], influenza, and hepatitis B vaccination, among others. In the general population, myocarditis is diagnosed in approximately 10–20 individuals per 100,000 per year [6], and occurs more commonly and at younger ages in males than females [7]. The highest reported incidence of myocarditis from vaccination occurred after a second dose of mRNA COVID-19 vaccine, and mostly among young men [8,9,10]. Most of these cases developed symptoms within the first week, typically 2–4 days post-vaccination. The prognosis for myocarditis patients varies according to etiology [11]. In the pre-COVID-19 era (1990–2018), among 620,195 reports filed in the Vaccine Adverse Event Reporting System (VAERS), 0.1% were attributable to myopericarditis; of these myopericarditis reports, 79% were in males [12]. However, the VAERS is primarily a safety signal detection and hypothesis-generating system, and cannot be used to determine whether the vaccine caused the adverse events [13]. Cardiovascular findings from mRNA COVID-19 vaccine included myocardial injury (myocarditis), coronary heart disease, heart failure, hypertension, and rhythm disorder [13]. The cardiovascular manifestations of COVID-19 infection include elevated cardiac biomarkers, myocarditis, cardiac arrhythmia, and venous thromboembolism [14].

Recently, a Centers for Disease Control and Prevention (CDC) advisory committee on immunization practices identified a likely association between the two COVID-19 mRNA vaccines from Pfizer–BioNTech and Moderna, and cases of myocarditis and pericarditis [15,16,17]. For the cardiovascular system, 4863 adverse events were reported among patients who received the BNT162b2 mRNA COVID-19 vaccine. Common findings observed with vaccines under study were tachycardia (16.41%), flushing (12.17%), increased heart rate (9.03%), hypertension (5.82%), and hypotension (3.6%) [13]. Although cardiovascular events have been reported with the COVID-19 vaccine, causality has yet to be established, because such cardiovascular adverse events are also common among the general public who do not receive the intervention [13]. Conducted during the implementation of the national Thai vaccination campaigns for adolescents, this study sought to characterize, classify, and evaluate the dynamics of cardiac function and electrocardiographic (ECG) abnormalities post-BNT162b2 mRNA COVID-19 vaccination to understand and identify cardiovascular effects that may predict cardiac complication by serial echocardiographic studies, ECG, and cardiac biomarkers for the early detection of subclinical myocarditis cases.

## 2. Materials and Methods

### 2.1. Study Design

This prospective cohort study focused on adolescent students from Kong Thabbok Upatham Changkol Kho So Tho Bo School and Wachirathamsatit School who received a second dose of the BNT162b2 mRNA COVID-19 vaccine. The study included subjects who were: (1) aged 13–18 years; (2) male or female; and (3) had received the first dose of the BNT162b2 mRNA COVID-19 vaccine without serious adverse event. Patients who had a history of cardiomyopathy, tuberculous pericarditis or constrictive pericarditis, and severe allergic reaction to the COVID-19 vaccine were excluded from the study. Laboratory tests included cardiac biomarkers (troponin-T, creatine kinase-myocardial band (CK-MB)), ECG, and echocardiography at three clinical visits (baseline, Day 3, Day 7, and Day 14 (optional for subjects with cardiac manifestation)) after receiving the second dose of the BNT162b2 mRNA COVID-19 vaccine. Participant data, including demographic data, clinical presentation, and laboratory findings, were recorded in a pre-defined case record form.

Potential subjects at Kong Thabbok Upatham Changkol Kho So Tho Bo School and Wachirathamsatit School were informed about the study by invitation letter, followed by an online meeting for parents and students to consider enrollment in the study. Informed consent documents were provided for interested parents to bring to the team investigator at enrollment. Enrollment was conducted from 3 November to 7 December 2021.

### 2.2. Diary Card

All participants received a diary card to record cardiac symptoms, such as chest pain or palpitations. Participants who developed cardiovascular effects or side effects from the vaccine could telephone the principal investigator and be transferred by phone to the medical team at the Hospital for Tropical Diseases for assessment. If the participant developed abnormal ECG, echocardiographic findings, or increased cardiac enzymes, the principal investigator scheduled patients for follow-up per protocol and for Day 14 lab assessments.

### 2.3. Definition of Cardiovascular Manifestation

In this study, cardiovascular manifestation was defined as one or more of the following:Chest pain/pericarditisDyspnea/orthopneaPalpitationHypertension/hypotensionTachycardia/bradycardiaShock/cardiogenic shockAbnormal ECG or abnormal rhythm or ECG changeBundle branch blockDecreased ejection fractionDiastolic dysfunctionElevation in at least one cardiac biomarker (troponin-T, CK-MB)/myocarditis

### 2.4. Definition of Myocarditis [18]

The diagnostic criteria for myocarditis were classified as either probable cases or confirmed cases. Myocarditis patients were those with the presence or worsening of more than one of the following clinical symptoms along with evidence of inflammation: (1) chest pain, pressure, or discomfort; (2) dyspnea, shortness of breath, or pain with breathing; (3) palpitation; or (4) syncope and more than one new finding of: (a) troponin level above upper normal limit of normal; (b) abnormal ECG or rhythm monitoring consistent with myocarditis; (c) abnormal cardiac function or wall motion on echocardiography; (d) cardiac magnetic resonance imaging (cMRI) findings consistent with myocarditis and no identifiable cause for symptoms and findings.

### 2.5. Definition of Pericarditis [18]

The diagnostic criteria for pericarditis included the new presence or worsening of more than two of the following clinical features: (1) acute chest pain; (2) pericardial rub on exam; (3) new ST-segment elevation or PR-segment depression on ECG; and (4) pericardial effusion on echocardiography or cMRI.

### 2.6. Cardiac Enzymes

High-sensitivity cardiac troponin-T assay (HS-cTnT) and CK-MB isoenzyme levels were determined for all participants at baseline, and on Day 3, Day 7, and Day 14 (optional) after the second vaccination dose. HS-cTnT was measured using the Elecsys troponin-T hs assay (Roche Diagnostics, Mannheim, Germany); serum levels > 14 ng/L were considered elevated. CK-MB was also measured by electrochemiluminescence immunoassay with Elecsys CK-MB (Roche Diagnostics, Mannheim, Germany); serum CK-MB levels > 6.22 ng/mL in males and >4.88 ng/mL in females were considered elevated. CRP levels > 5 mg/L were considered elevated. ESR > 20 mm/h was considered elevated. All tests during the study were performed by hospital technicians.

### 2.7. Echocardiography Protocol

Echocardiograms during the examination were recorded using the Vivid E9 ultrasound platform (GE Healthcare, Chicago, IL, USA). All echocardiographic images were recorded and reviewed by two cardiologists. Routine two-dimensional echocardiograms and color-flow Doppler images were obtained in the standard parasternal long axis view, subcostal view, and apical four-chamber views. The left ventricular walls and dimensions were measured in accordance with the guidelines of the American Society of Cardiology. Transmitral pulsed-wave Doppler velocities (peak E- and A-wave velocities) were measured in the apical four-chamber view with the sample volume positioned at the mitral valve.

Pericardial effusion and other anatomical and functional findings were recorded when present. Echocardiographic studies were performed at baseline, and on Day 3, Day 7, and Day 14 (optional) after the second BNT162b2 mRNA COVID-19 vaccine dose.

### 2.8. Sample Size Calculation

To calculate estimated sample size, we used the estimated prevalence described in a previous study, where the incidence of cardiac manifestation among patients who received COVID-19 vaccine was found to be 4.6% [19]. Based on a population size of 1000 students after the first BNT162b2 mRNA COVID-19 vaccine dose, a minimum sample size of 297 was calculated to be sufficient for this study to determine cardiovascular effects, with an error of 2% at a 95% confidence interval. We expected that 5% of patients could be lost to follow-up or drop out before study end.

### 2.9. Statistical Analysis

All data were analyzed using SPSS version 18 (IBM Corp., Armonk, NY, USA). Categorical variables were summarized and expressed as frequencies and percentages. Quantitative variables were presented as mean ± SD. The chi-square test or Fisher’s exact test was used to assess differences between groups, as appropriate. For all analyses, *p* < 0.05 was considered statistically significant.

## 3. Results

A total of 314 participants were enrolled into the study; 13 of these were excluded from analysis, being lost to follow-up. The remaining 301 participants in the study made up the analysis set, as shown in Figure 1.

### 3.1. Characteristics of the Study Population

The mean age of the participants was 15 years (standard deviation (SD) 1.6 years; range 13–18 years). Of the 301 participants, 202 (67.1%) were male. All participants were healthy and without any abnormal symptoms before receiving the second dose of vaccine. The majority of the participants (257/301, 85.38%) had no underlying diseases. There were 44 participants with underlying medical conditions, including allergic rhinitis, asthma, thalassemia trait, and G6PD deficiency. There was no significant difference in clinical characteristics between the 13–15-year-old group and the older adolescents in this study cohort. The clinical characteristics of all 301 participants are shown in Table 1.

During the follow-up period, after receiving the second dose of vaccine, two patients were hospitalized and one patient was supervised in the ICU during hospitalization, mainly for observation of arrhythmia. The mean length of stay in the hospital was 4.5 days (range 2–7). None of the participants died, required mechanical ventilation, or required inotropic support. 

### 3.2. Cardiovascular Findings

Cardiovascular adverse events observed during the study were tachycardia (7.64%), shortness of breath (6.64%), palpitation (4.32%), chest pain (4.32%), and hypertension (3.99%). Fifty-four patients had abnormal electrocardiograms (predominantly sinus tachycardia or sinus arrhythmia) after vaccination. All 54 of these patients had normal left ventricular ejection fraction. Three patients had minimal pericardial effusion. The cMRI revealed findings compatible with subacute myopericarditis (no evidence of myocardial edema with evidence of nonischemic delayed enhancement at lateral wall and pericardial enhancement at inferolateral wall) in one patient who was diagnosed with myopericarditis, as shown in Figure 2A–C. In addition, follow-up cMRI 5 months later showed no evidence of myocardial edema, myocardial delayed enhancement, or myocardial fibrosis. There is evidence of resolved myocarditis, as shown in Figure 2D–F. Incidental findings on echocardiography included a bicuspid aortic valve and a dilated coronary sinus from persistent left superior vena cava. Six patients had mitral valve prolapse, and six patients had hypertension (HTN). Three patients diagnosed with myopericarditis and pericarditis were treated with nonsteroidal anti-inflammatory drugs (NSAIDs) for 2 weeks with no residual symptoms and complete follow-up. The patients presenting with myopericarditis, subclinical myocarditis, and pericarditis are shown in Table 2.

### 3.3. Evaluation of Patients with Elevated Biomarkers or Positive Lab Assessments 

Seven patients had elevated biomarkers or positive lab assessments. The most commonly presented symptom was chest pain, followed by chest discomfort, fever, and headache. Three patients aged 13–18 years, who presented with chest pain and biomarker elevation, were evaluated; all three presented 24–48 h after receiving the second dose of the vaccine. Four patients had no symptoms but elevated cTnT (peak level 15.44–38.68 ng/L; normal level < 14 ng/L). The characteristics of the patients with elevated biomarkers or positive lab assessments are shown in Table 3. All patients were male and had abnormal electrocardiograms, particularly sinus tachycardia. The clinical course was mild in all patients.

### 3.4. Evaluation of Patients Developing Abnormal ECG Post-Vaccination

After vaccination, ECG revealed that of the 301 patients, 247 (82.06%) had normal sinus rhythm, while an abnormal ECG finding was noted in 54 patients (17.94%) (Table 4). The most common abnormal ECG finding was sinus rhythm with sinus arrhythmia (7.31%), followed by sinus tachycardia (6.64%) and sinus bradycardia (1.33%). Of the two patients with abnormal rhythm, one had junctional escape rhythm, and one had ectopic atrial rhythm. Arrhythmia was observed as premature ventricular contractions in two patients (0.66%), and three (1%) had premature atrial contraction. One case (0.33%) had diffused ST elevation with PR depression. 

### 3.5. Evaluation of Patients with Serial Echocardiographic Findings

All participants underwent a follow-up echocardiography examination at baseline, Day 3, Day 7, and Day 14 (optional). There were no significant differences between the echocardiographic findings by study day, as shown in Table 5.

## 4. Discussion

This prospective cohort study focuses on cardiovascular manifestations after BNT162b2 mRNA vaccination. Immunization against COVID-19 infection using mRNA-based vaccines is a new technology [20]. On 10 May 2021, the US Food and Drug Administration (USFDA) expanded the use of the Pfizer–BioNTech vaccine to include adolescents aged 12–15 years [21]. In the COVID-19 era, the risks of myocarditis after an mRNA vaccine injection, especially in male adolescents, have raised particular concerns. In July 2021, the CDC reported an association between COVID-19 mRNA vaccines and suspected cases of myocarditis and pericarditis. The incidence rate of myocarditis/pericarditis after mRNA COVID-19 vaccine was reported to be as low as 12.6 cases per million second dose mRNA vaccines among those aged 12–39 years [8,15]. In contrast, our study found one case of myopericarditis, four cases of subclinical myocarditis, and two cases of pericarditis among 301 participants, and each case had mild symptoms. The incidence of myocarditis/pericarditis found in our study may be higher than the other studies due to the study protocol, which required determining baseline troponin-T, CK-MB, ECG, and echocardiography before vaccination. Two retrospective studies from Israel [8,9] showed a slightly different incidence compared with CDC data, possibly resulting from different data collection methods and different criteria for diagnosing myocarditis. Montgomery and colleagues reported on 23 male military personnel diagnosed with myocarditis after presenting with acute sudden onset of chest pain within 4 days after mRNA COVID-19 vaccine [22]. Another prospective study reported six males who were hospitalized with suspected myocarditis, all shortly after a second dose of BNT162b2 mRNA COVID-19 vaccine [23].

The potential mechanism of mRNA COVID-19 vaccine-induced myocarditis remains unknown. It has been suggested that excessive innate immune activation by both lipid nanoparticle and RNA components of COVID-19 vaccines can cause myocarditis [24,25]. Endosomal toll-like receptor (TLR) TLR3, TLR7, and TLR8 in immune cells and RIG-I and MDA5 in nonimmune cells act as a natural defense against foreign RNA but can cross-react with in vitro transcribed (IVT) RNA [26]. The activation of these receptors triggers an inflammatory cascade, assembly of inflammasome platform production of type I interferons, and nuclear translocation of NK-kB [24,25,27]. Inflammasomes are large multiproteins that are responsive to pathogen- and stress-associated cellular insults. Inflammasomes lead to the secretion of the pleiotropic IL-1 family (IL-1β and IL-18), and pyroptosis [28]. In one case report, patients with vaccine-induced myocarditis had increased levels of interleukin 1 (IL-1) receptor antagonist, interleukin 5 (IL-5), and interleukin 16 (IL-16) [5]. Myocarditis associated with COVID-19 mRNA vaccination mainly occurs in male teenagers within 1 week post-vaccination, typically the second dose vaccination, with recovery of cardiac function within 1–5 weeks after hospitalization [8,12]. The mechanism is unknown, but may be related to the mRNA sequence that encodes for the spike protein of SARS-CoV-2, or the immune response following vaccination [18,29]. By contrast, the incidence of COVID-19-associated cardiac injury or myocarditis is much higher, estimated to be 100 times higher than mRNA COVID-19-related myocarditis [30,31]. Moreover, mRNA vaccine-related myocarditis is characterized by overall mild presentation and favorable outcomes. In our study, chest pain was considered an alarming side effect after BNT162b2 mRNA vaccine injection. Although clinical symptoms spontaneously resolved rapidly in all patients, the potential for cardiac fibrosis vaccine-related myocarditis remains unknown. The long-term outcomes of COVID-19 vaccine have not been described, but in our study nearly 100% of patients with symptoms had recovered within 1–2 weeks, concordant with another study [18]. Long-term surveillance with follow-up cardiac imaging, especially cardiac MRI in patients with vaccine-related myocarditis, is required.

In this study, one patient who was admitted to the ICU Department with negative PCR for COVID-19, was diagnosed with myopericarditis after mRNA COVID-19 vaccine, and showed abnormalities in cardiac enzymes (troponin-T, CK-MB), ECG, echocardiography, and cardiac MRI. As our patient remained clinically stable, endomyocardial biopsy was not indicated during hospitalization. He was treated with ibuprofen for 2 weeks without guideline-directed medical therapy for heart failure because his imaging showed normal LVEF. Treatment of vaccine-induced myocarditis includes corticosteroids, NSAIDs, colchicine, and, in severe cases, IVIG [32,33]. Corticosteroids have been proposed for the treatment of vaccine-induced myocarditis [33]. Nonpharmacological strategies to prevent cardiovascular side effects of COVID-19 vaccination include oral and systemic administration of ascorbic acid. An observational study has found that low serum ascorbic acid can increase cardiovascular disease in humans [34]. Ascorbic acid can prevent cardiotoxic events and can reduce the relative risk for cardiovascular events that could reduce inflammation related to cardiovascular events after COVID-19 vaccination [35]. All patients with subclinical myocarditis had elevated troponin-T without CK-MB elevation; troponin-T level was also highly sensitive for post-vaccination screening for myocarditis. Two patients diagnosed with pericarditis had normal troponin-T levels, but generally raised c-reactive protein (CRP) or erythrocyte sedimentation rate (ESR). All seven patients with elevated biomarker levels or positive lab assessments showed normal left ventricular ejection fraction (LVEF), and only three patients had minimal pericardial effusion. Conventional echocardiography may not be the ideal diagnostic tool in suspected vaccine-induced myocarditis because of its mild clinical symptoms; conventional echocardiography appeared normal in these patients. Speckle tracking echocardiography (STE) strain and strain-rate parameters are useful diagnostic measures with high sensitivity for the early detection of subclinical ventricular dysfunction [36], and cardiac MRI may be the best way to confirm a diagnosis of myocarditis in the majority of cases. In this study, the disease course was mild in all of our patients with cardiac symptoms; they were treated with ibuprofen for 2 weeks and cardiac enzymes returned to normal after 1–2 weeks of outpatient treatment. Myocarditis may have more severe clinical manifestations requiring inotropic drug or mechanical support; however, one patient with myopericarditis in our study follow-up with cMRI at 5 months post-vaccination showed complete recovery and no scar. If an adolescent presents with myopericarditis after COVID-19 mRNA vaccination, a booster shot is contraindicated.

This study had some limitations. Due to the national Thai vaccination campaign against the COVID-19 pandemic, the government declared the timing for the first mRNA COVID-19 vaccine dose urgently, so there was limited time for the Ethics Committee (EC) to process the first dose; however, myocarditis/pericarditis was more common with the second mRNA COVID-19 vaccination dose. Our study included some participants with subclinical myocarditis presenting without symptom other than elevated troponin-T. It was necessary to request parental permission for blood testing on Day 14, which may have limited participation.

Strengths: this is the first prospective study in Thailand during the national campaign of vaccination against the COVID-19 pandemic for adolescents. The strengths of our study include its prospective design employing serial cardiac enzyme, ECG, and echocardiographic measurements at three separate visits (baseline, Day 3, and Day 7). Two cardiologists at different institutions worked together to confirm the diagnoses of myocarditis, myopericarditis, and pericarditis. Another strength of this study is that participants and parents were able to contact the principal investigators directly online or by telephone for consultation and immediate treatment.

## 5. Conclusions

In this observational study, clinically suspected myopericarditis was temporarily associated with the BNT162b2 mRNA COVID-19 vaccine in a small proportion of adolescent patients. Chest pain is an alarming symptom in patients receiving BNT162b2 mRNA COVID-19 vaccination, especially a second dose of BNT162b2. The risk for these symptoms was found to be higher than reported elsewhere. The adverse cardiovascular manifestations observed in this adolescent cohort were both mild and transient. 

## Figures and Tables

**Figure 1 tropicalmed-07-00196-f001:**
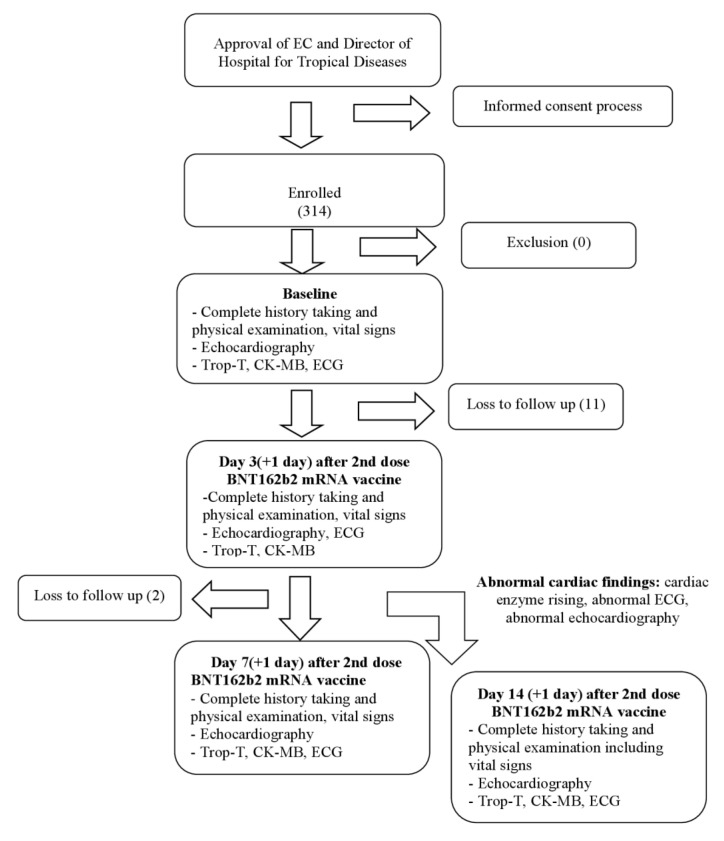
Study flow chart. CK-MB, creatine kinase-myocardial band; ECG, electrocardiography; Trop-T, troponin-T.

**Figure 2 tropicalmed-07-00196-f002:**
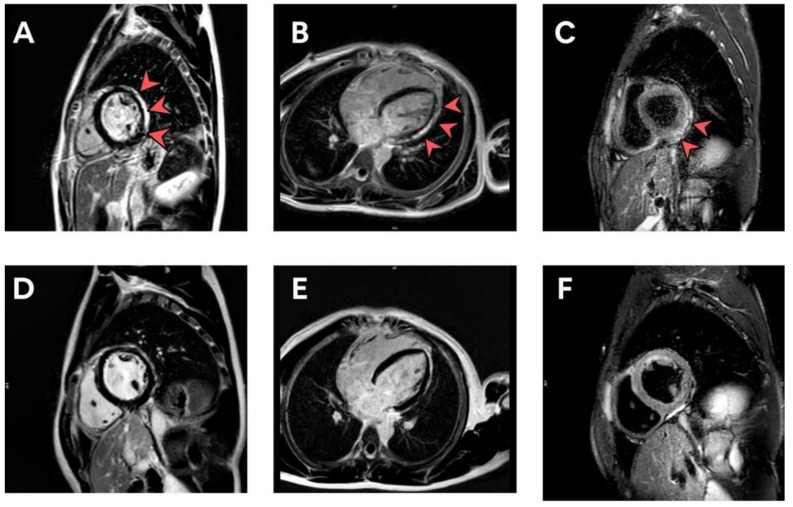
(**A**–**F**) cMRI illustrating LGE in a patient with subacute myopericarditis at the time of diagnosis (**A**–**C**) and 5 months post-diagnosis (**D**–**F**). cMRI, cardiac magnetic resonance imaging; LGE, late gadolinium enhancement.

**Table 1 tropicalmed-07-00196-t001:** Clinical characteristics of the 301 adolescents after the second COVID-19 vaccination.

Characteristic	Overall (*n* = 301)	13–15 y (*n* = 207)	16–18 y (*n* = 94)	*p*-Value
Age, y	15 ± 1.6	14 ± 0.8	17 ± 0.7	-
BMI (kg/m^2^)	21 ± 5.0	20 ± 4.8	22 ± 5.2	0.017
Male sex, *n* (%)	202 (67.1)	110 (53.1)	92 (97.9)	<0.0001 *
Underlying disease *n* (%)	44 (14.6)	31 (15.0)	13 (13.8)	0.795
Allergic rhinitis	24 (8.0)	17 (8.2)	7 (7.4)	0.813
Asthma	7 (2.3)	5 (2.4)	2 (2.1)	0.869
Thalassemia trait	5 (1.7)	3 (1.4)	2 (2.1)	0.688
G6PD deficiency	4 (1.3)	3 (1.4)	1 (1.1)	0.782
Attention deficit	1 (0.3)	1 (0.5)	0	0.500
Epilepsy	1 (0.3)	1 (0.5)	0	0.500
Migraine	1 (0.3)	1 (0.5)	0	0.500
Thyrotoxicosis	1 (0.3)	0	1 (1.1)	0.500
Symptoms, *n* (%)				
Fever	50 (16.6)	30 (14.5)	20 (21.3)	0.093
Palpitation	12 (4.0)	10 (4.8)	2 (2.1)	0.268
Chest pain	8 (2.7)	5 (2.4)	3 (3.2)	0.699
Shortness of breath	19 (6.3)	16 (7.7)	3 (3.2)	0.134
Headache	35 (11.6)	27 (13.0)	8 (8.5)	0.257
Laboratory findings				
Troponin-T, ng/L	5.6 ± 2.5	5.4 ± 2.5	5.9 ± 2.5	0.112
CK-MB ng/mL	1.4 ± 0.9	1.4 ± 0.9	1.5 ± 0.9	0.473
Treatment and hospital course				
NSAIDS, *n* (%)	3 (1.0)	1 (0.5)	2 (2.1)	0.178
Hospitalization, *n* (%)	2 (0.7)	0	2 (2.1)	0.035
ICU admission, *n* (%)	1 (0.3)	0	1 (1.1)	0.138

* Statistically significant (chi-square test). BMI, body mass index; CK-MB, creatine kinase-myocardial band; NSAIDS, nonsteroidal anti-inflammatory drugs; Trop-T, Troponin-T.

**Table 2 tropicalmed-07-00196-t002:** Presentation with myopericarditis, subclinical myocarditis, and pericarditis after second dose vaccination.

Variable	Value
**Presenting symptoms and signs—Number/total number (%)** Chest pain Chest discomfort Pericardial effusion Fever Headache Palpitation Dyspnea	3/7 (42.86) 3/7 (42.86) 3/7 (42.86) 4/7 (57.14) 2/7 (28.57) 1/7 (14.29) 1/7 (14.29)
**Vital signs on day of symptoms (Mean ± SD)**	
Temperature—°C Blood pressure—mmHg Systolic Diastolic Heart rate—beats/min	36.4 ± 0.4 114.9 ± 10.9 70.7 ± 7.8 92.71 ± 21.3
Shock-Number/total number. (%)	0/7 (0)
**Electrocardiographic findings—Number/total number (%)** Normal sinus rhythm Sinus rhythm with sinus arrhythmia Diffuse ST elevation with PR depression Sinus arrhythmia with PAC Sinus tachycardia Junctional escape rhythm	1/7 (14.29) 2/7 (28.57) 1/7 (14.29) 1/7 (14.29) 1/7 (14.29) 1/7 (14.29)
**Laboratory values** Elevated troponin-T	5/7 (71.43)
**Clinical course** Arrhythmia ICU admission Need for inotrope or vasopressor Death	4/7 (57.14) 1/7 (14.29) 0/7 (0) 0/7 (0)
**Treatment and hospital course** Ibuprofen (NSAIDs)	3/7 (42.86)

Data are reported as percentage (%) and means ± standard deviations; °C, degree Celsius; NSAIDs, nonsteroidal anti-inflammatory drugs; PAC, premature atrial contraction.

**Table 3 tropicalmed-07-00196-t003:** Characteristic of patients with elevated biomarker levels or positive lab assessments.

Demographic				Clinical Presentation										Echocardiography				
No.	Age (y)	Sex	Classification	Peak CRP (mg/ L)	Peak ESR (mm/hr)	CK-MB Level (ng/mL)				Troponin-T (pg/mL)				LVEF%				Pericardial Effusion
						1	2	3	4	1	2	3	4	1	2	3	4	
1	16	Male	Myopericarditis	86.6	19	1.25	109.6	2.36	1.67	3.18	593	37.2	10.9	75.3	73.7	77.2	84.7	Yes
2	15	Male	Pericarditis	1.3	7	1.11	1.34	1.52	1.46	2.58	3.77	6.04	3.93	61.5	60.2	74.1	70.7	Yes
3	17	Male	Pericarditis	10.5	8	1.99	1.87	1.72	2.71	4.54	8.03	7.87	6.75	78.9	77.5	61.0	67.2	Yes
4	13	Male	Subclinical myocarditis	0.3	-	1.39	1.72	2.28	-	8.56	10.3	34.94	-	58.6	59.2	75.4	-	No
5	14	Male	Subclinical myocarditis	0.5	-	3.00	2.06	3.06	-	3.73	28.6	38.68	-	79.6	60.1	76.2	-	No
6	13	Male	Subclinical myocarditis	0.9	-	3.90	3.67	5.10	-	5.35	14.87	16.81	-	64.3	76.2	78.9	-	No
7	17	Male	Subclinical myocarditis	4.3	-	2.25	2.32	2.41	-	3.12	13.06	15.44	-	70.8	52.4	53.8	-	No

CRP, C-reactive protein; ESR, erythrocyte sedimentation rate; CK-MB, creatine kinase-myocardial band; LVEF, left ventricular ejection fraction.

**Table 4 tropicalmed-07-00196-t004:** Electrocardiographic findings after second vaccine dose.

Rhythm	Number (*n* = 301)
Normal sinus rhythm	247 (82.06%)
Sinus rhythm with sinus arrhythmia	22 (7.31%)
Sinus tachycardia	20 (6.64%)
Sinus bradycardia	4 (1.33%)
Premature atrial contraction (PAC)	3 (1%)
Premature ventricular contraction (PVC)	2 (0.66%)
Junctional escape rhythm	1 (0.33%)
Ectopic atrial rhythm	1 (0.33%)
Diffuse ST elevation with PR depression	1 (0.33%)

Data are reported as percentage (%). PAC, premature atrial contraction; PVC, premature ventricular contraction.

**Table 5 tropicalmed-07-00196-t005:** Comparison between cardiac function on day of baseline (D0), Day 3 after the second vaccine dose (D3), and Day 7 after the second vaccine dose (D7).

Cardiac Function	Day0	Day3	Day7	*p*-Value D0 vs. D3	*p*-Value D0 vs. D7	*p*-Value D3 vs. D7
IVSD, mean ± SD	1.18 ± 4.83	0.99 ± 1.51	0.93 ± 1.32	0.508	0.383	0.596
LVIDd, mean ± SD	4.43 ± 3.22	4.34 ± 3.02	4.81 ± 5.02	0.735	0.209	0.168
LVPWd, mean ± SD	0.96 ± 0.67	0.91 ± 0.81	1.28 ± 5.16	0.431	0.299	0.234
LVIDs, mean ± SD	2.64 ± 1.65	2.85 ± 2.82	2.60 ± 0.5	0.250	0.730	0.128
LVEF, mean ± SD	68.68 ± 9.27	68.21 ± 9.18	68.30 ± 8.56	0.490	0.585	0.878
MV flow E-wave velocity mean ± SD	99.32 ± 18.77	98.98 ± 20.47	99.93 ± 21.05	0.791	0.773	0.391
MV flow A-wave velocity mean ± SD	53.37 ± 16.10	52.39 ± 15.39	51.23 ± 16.18	0.432	0.046	0.217
MV annulus E-wave velocity, mean ± SD	2.11 ± 1.28	2.03 ± 0.63	2.06 ± 1.18	0.281	0.632	0.653
e’, mean ± SD	0.12 ± 0.05	0.13 ± 0.10	0.18 ± 0.55	0.227	0.097	0.065
E/e´, mean ± SD	8.33 ± 2.63	8.22 ± 2.03	9.29 ± 8.27	0.551	0.277	0.125
a´, mean ± SD	0.08 ± 0.06	0.08 ± 0.04	0.16 ± 0.79	0.393	0.201	0.091

Data are presented as means ± standard deviations. *p*-values correspond to paired *t*-tests. a´, late (atrial) diastolic mitral annular velocity; e´, early diastolic mitral annular velocity; E/e´, ratio of peak early mitral inflow velocity to early diastolic mitral annular velocity; IVSD, interventricular septal end diastole; LVEF, left ventricular ejection fraction; LVIDS, left ventricular internal diameter end systole; LVIDd, left ventricular internal diameter end diastole; LVPWd, left ventricular posterior wall end diastole; MV, mitral valve.

## Data Availability

The dataset can be requested from the corresponding author.

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
