# Peer review of "Cardiovascular Manifestation of the BNT162b2 mRNA COVID-19 Vaccine in Adolescents"

_tropicalmed, 2022, doi:10.3390/tropicalmed7080196_

Round 1

Reviewer 1 Report

The authors investigate the CV effects of one of the COVID-19 vaccines. The study highlights an interesting and hot topic of this current historical moment. The study is easy to read and set to a broad audience. Before publication there are critical aspects to address:

-The authors miss use medical terminology. Cardiovascular effects is not synonym of patients symptoms or isolated findings. This must be addressed through the manuscript. 

- A paragraph addressing incidence of CV findings and vaccine compared to COVID-19 CV findings should be added

-Conclusion and abstract must be augmented. The message should be that vaccines are safe and that all adverse effects were temporary and that there is no comparison with COVID-19 complications. 

-Limitations are not described in the article

Author Response

Response to Reviewer #1:

We are delighted that you think our work is helpful. Thank you for taking the time and energy to help us improve the paper. Below are our responses to your comments.

The authors investigate the CV effects of one of the COVID-19 vaccines. The study highlights an interesting and hot topic of this current historical moment. The study is easy to read and set to a broad audience. Before publication there are critical aspects to address:

  1. COMMENT:

The authors miss use medical terminology. Cardiovascular effects is not synonym of patients or isolated findings. This must be addressed through the manuscript.

RESPONSE: Thank you for your valuable comments. We apologize for the medical terminology. We wish to change “cardiovascular effects” to “cardiovascular manifestation(s)” instead.

  1. COMMENT:

A paragraph addressing incidence of CV findings and vaccine compared to COVID-19 CV findings should be added.

RESPONSE: Thank you very much for your valuable suggestion. We have added CV findings to the introduction section (lines 55-59, page 2 in the manuscript).

3. COMMENT:

Conclusion and abstract must be augmented. The message should be that vaccines are safe and that all adverse effects were temporary and that there is no comparison with COVID-19 complications.

RESPONSE: Thank you very much for your important comment. We have augmented the conclusion and abstract as suggested; Abstract (lines 27-31, page 1 in the manuscript) and Conclusion (lines 360-366, page 13 in the manuscript). In my opinion, I report findings of post-vaccination cardiovascular manifestations among a small and specific cohort. I think due to objects of this study, I couldn’t say vaccines are safe or unsafe. I could say only all of the cardiovascular manifestations were both mild and transient with full recovery.

 4. COMMENT:

Limitation are not described in the article.

RESPONSE: The limitations are described in lines 344-351, page 12 of the manuscript.

Reviewer 2 Report

Manuscript titled " Cardiovascular Effects of the BNT162b2 mRNA COVID-19 Vaccine in Adolescents" is a very interesting ,mauscript describing the cardivoascular side effects of COVID-19vaccination in adolescents. Overall structure is of good quality, references are updated on this topic, methods are clear. However, authors should improve the manuscript in several parts:

1) in discussion, authors should add pharmacological  strategies aimed to prevent cardiovascular side effects of COVID-19 vaccination like glucocorticoids or selective cytokine inhibitors.

2) in discussion, authors should add also non-pharmacological  strategies aimed to prevent cardiovascular side effects of COVID-19 vaccination like oral and systemic administration of ascorbic acid ( cite 10.3390/antiox9121182 ) or resveratrol or arginin or medicinal mushrooms ( cite 10.18632/oncotarget.24984) that could reduce inflammation-related to cardiovascular events in thee patients.

3)please, add more data on the role of inflammasome and interleukin 1 and 6 in pathogenesis of COVID-19 vaccination-related cardiovascular events.

Author Response

Response to Reviewer #2:

We are delighted that you think our work is helpful. Thank you for taking the time and energy to help us improve the paper. Below are our responses to your comments.

Manuscript titled “Cardiovascular Effects of the BNT162b2 mRNA COVID-19 Vaccine in Adolescents” is a very interesting, manuscript describing the cardiovascular side effects of COVID-19 vaccination in adolescents. Overall structure is of good quality, references are updated on this topic, methods are clear. However, authors should improve the manuscript in several parts:

  1. COMMENT: in discussion, authors should add pharmacological strategies aimed to prevent cardiovascular side effects of COVID-19 vaccination like glucocorticoids or selective cytokines inhibitors

 RESPONSE: Thank you for your valuable comments. We have been unable to find pharmacological treatments, such as steroids, to prevent cardiovascular side-effects. In clinical practice, physicians should not prescribe corticosteroids to prevent disease, but use them only for treatment. We have added corticosteroids for the treatment of vaccine-induced myocarditis, especially for severe cases. We have added lines 315-317 on page 12 of the manuscript.

  1. COMMENT: in discussion, authors should add also non-pharmacological strategies aimed to prevent cardiovascular side effects of COVID-19 vaccination like oral and systemic administration of ascorbic acid (cite 10.3390/antiox9121182) or resveratrol or arginin or medical mushrooms (cite 10.18632/oncotarget.24984) that could reduce inflammation-related to cardiovascular events in these patients.

RESPONSE: Thank you for your valuable comments. We have added ascorbic acid to prevent cardiovascular side effects from vaccine. In my opinion, ascorbic acid is an antioxidant, decreasing oxidative stress; most of the general population used ascorbic acid. We have added lines 317-323 on page 12 of the manuscript.

  1. COMMENT: please, add more data on the role of inflammasome and interleukin 1 and 6 in pathogenesis of COVID-19 vaccination-related cardiovascular events.

 RESPONSE: Thank you for your important points. We have added lines 282-293 on page 11 of the manuscript, as suggested.

Reviewer 3 Report

This is a prospective study on cardiovascular effects of BNT162b2 mRNA COVID-19 vaccine in adolescents. This article is well-written; however, the following points should be addressed:

Line 22: “ranging from tachycardia, palpitation, and myopericarditis” is probably a typo of “ranging from tachycardia or palpitation to perimyocarditis”.

Line 27-28: “Hence, adolescents receiving mRNA vaccines should be monitored for side effects” should be revised to “.......monitored for cardiovascular side effects.”

Line 99 – 109: Symptoms (such as palpitation), disease names (such as cardiogenic shock), and abnormal test findings (such as decreased ejection fraction) are randomly listed. Please categorize them in appropriate style and list these items separately.

Line 112-113: “Myocarditis patients were those with the presence or worsening of more than one of the following clinical symptoms along with the evidence of inflammation:”

Line 188: does “palpitation” include PVCs and PACs? If so, it would be better to clarify.

Table 2: what does “signs-no/total no” mean? If “no” means “numbers”, please spell it out.

In Discussion, line 312-313: “COVID-19 mRNA vaccination has an extremely favorable outcome and should be recommended for adolescents.” This sentence sounds different from what the authors have described. Myocarditis is sometimes fatal and its mortality rate is reported to be more than 20%. If an adolescent presented with myocarditis after COVID-19 mRNA vaccination, the booster shot is definitely not recommended.

This prospective research is very valuable and informative to the healthcare providers worldwide.

Author Response

Response to Reviewer #3:

We are delighted that you think our work is helpful. Thank you for taking the time and energy to help us improve the paper. Below are our responses to your comments.

This is a prospective study on cardiovascular effects of the BNT162b2 mRNA COVID-19 vaccine in adolescents. This article is well-written; however, the following points should be addressed.

  1. COMMENT: Line 22: “ranging from tachycardia, palpitation, and myopericarditis” is probably a typo of “ranging from tachycardia or palpitation to perimyocarditis”.

RESPONSE: Thank you for your comments. We have changed lines 23-24            on page 1 of the manuscript.

  1. COMMENT: Line 27-28 (30-31): “Hence, adolescent receiving mRNA vaccines should be monitored for side effects” Should be revised to “… monitored for cardiovascular side effects”

RESPONSE: Thank you for your comments. We have changed line 30-31 on page 1 of the manuscript accordingly.

  1. COMMENT: Line 99-109 (104-117): Symptoms (such as palpitation), disease names (such as cardiogenic shock), and abnormal test findings (such as decreased ejection fraction) are randomly listed. Please categorize them in appropriate style and list these items separately.

RESPONSE: Thank you for your comments. We have made the suggested changes in lines 104-117 on page 3 of the manuscript.

  1. COMMENT: Line 112-113 (120-121): “Myocarditis patients were those with the presence or worsening of more than one of the following clinical symptoms along with the evidence of inflammation:”

RESPONSE: Thank you for your comments. We have changed line 121 on page 3 of the manuscript accordingly.

  1. COMMENT: Line 188 (196): does “palpitation” include PVCs and PACs? If so, it would be better to clarify.

RESPONSE: Thank you for your valuable comments. In line 196, among the participants who presented with palpitation ECG ranging from normal sinus rhythm, sinus tachycardia, sinus arrhythmia, only one patient showed PVC. The most common ECG among palpitation patients was sinus tachycardia.

  1. COMMENT: Table 2: What does “signs-no/total no” mean? If “no” means “numbers”, please spell it out.

RESPONSE: Thank you for your suggestion. We have changed “no” to “number” for greater clarity. At table 2, page 7 in the manuscript.

  1. COMMENT: In discussion, line 312-313 (340-341): “COVID-19 mRNA vaccination has an extremely favorable outcome and should be recommended for adolescents. “This sentence sounds different from what the authors have described. Myocarditis is sometimes fatal and its mortality rate is reported to be more than 20%. If an adolescent presented with myocarditis after COVID-19 mRNA vaccination, the booster shot is definitely not recommended.

RESPONSE: Thank you for your valuable suggestion. We agree with your comment that myocarditis can be fatal. We have deleted this sentence from the manuscript. Also, I added “If an adolescent presents with myopericarditis after COVID-19 mRNA vaccination, a booster shot is contraindicated”, line 342-343 on page 12 of the manuscript.

Round 2

Reviewer 1 Report

The authors have made the required adjustments. The article is now fit for publication.